# Role of Body Mass and Physical Activity in Autonomic Function Modulation on Post-COVID-19 Condition: An Observational Subanalysis of Fit-COVID Study

**DOI:** 10.3390/ijerph19042457

**Published:** 2022-02-21

**Authors:** Ana Paula Coelho Figueira Freire, Fabio Santos Lira, Ana Elisa von Ah Morano, Telmo Pereira, Manuel-João Coelho-E-Silva, Armando Caseiro, Diego Giulliano Destro Christofaro, Osmar Marchioto Júnior, Gilson Pires Dorneles, Luciele Guerra Minuzzi, Ricardo Aurino Pinho, Bruna Spolador de Alencar Silva

**Affiliations:** 1Department of Health Sciences, Central Washington University Ellensburg, Ellensburg, WA 98926, USA; 2Physiotherapy Department, Universidade do Oeste Paulista (UNOESTE), Presidente Prudente 19050-920, Brazil; 3Exercise and Immunometabolism Research Group, Postgraduate Program in Movement Sciences, Department of Physical Education, Universidade Estadual Paulista (UNESP), Presidente Prudente 19060-900, Brazil; anaelisavhm@yahoo.com.br (A.E.v.A.M.); marchioto@hotmail.com (O.M.J.); lucielegm@gmail.com (L.G.M.); brunaspolador@gmail.com (B.S.d.A.S.); 4Polytechnic of Coimbra, ESTESC, Laboratory Biomedical Sciences, 3046-854 Coimbra, Portugal; telmo@estescoimbra.pt (T.P.); armandocaseiro@estescoimbra.pt (A.C.); 5Molecular Physical-Chemistry R & D Unit, Faculty of Science and Technology, University of Coimbra, 3004-535 Coimbra, Portugal; 6Laboratory for Applied Health Research (LabinSaúde), 3046-854 Coimbra, Portugal; 7Faculty of Sport Science and Physiucal Education, University of Coimbra, CIDAF, 3000-456 Coimbra, Portugal; mjcesilva@hotmail.com; 8Postgraduate Program in Movement Sciences, Department of Physical Education, Universidade Estadual Paulista (UNESP), Presidente Prudente 19060-900, Brazil; diego.christofaro@unesp.br; 9Cellular and Molecular Immunology Laboratory, Universidade Federal de Ciências da Saúde de Porto Alegre, Porto Alegre 90050-170, Brazil; gilsondorneles@gmail.com; 10Graduate Program in Health Sciences, School of Medicine, Pontificia Universidade Catolica do Parana, Curitiba 80215-901, Brazil; rapinho12@gmail.com

**Keywords:** COVID-19, autonomic nervous system, heart rate, obesity, exercise

## Abstract

The harmful effects of coronavirus disease 2019 (COVID-19) can reach the autonomic nervous system (ANS) and endothelial function. Therefore, the detrimental multiorgan effects of COVID-19 could be induced by deregulations in ANS that may persist after the acute SARS-CoV-2 infection. Additionally, investigating the differences in ANS response in overweight/obese, and physically inactive participants who had COVID-19 compared to those who did not have the disease is necessary. The aim of the study was to analyze the autonomic function of young adults after mild-to-moderate infection with SARS-CoV-2 and to assess whether body mass index (BMI) and levels of physical activity modulates autonomic function in participants with and without COVID-19. Patients previously infected with SARS-CoV-2 and healthy controls were recruited for this cross-sectional observational study. A general anamnesis was taken, and BMI and physical activity levels were assessed. The ANS was evaluated through heart rate variability. A total of 57 subjects were evaluated. Sympathetic nervous system activity in the post-COVID-19 group was increased (stress index; *p* = 0.0273). They also presented lower values of parasympathetic activity (*p* < 0.05). Overweight/obese subjects in the post-COVID-19 group presented significantly lower parasympathetic activity and reduced global variability compared to non-obese in control group (*p* < 0.05). Physically inactive subjects in the post-COVID-19 group presented significantly higher sympathetic activity than active subjects in the control group. Parasympathetic activity was significantly increased in physically active subjects in the control group compared to the physically inactive post-COVID-19 group (*p* < 0.05). COVID-19 promotes changes in the ANS of young adults, and these changes are modulated by overweight/obesity and physical activity levels.

## 1. Introduction

Coronavirus disease (COVID-19) is a recognized infectious disease caused by SARS-CoV-2 infection that has rapidly spread worldwide, reaching a pandemic level in 2020. COVID-19 was reported in over 170 countries and has brought unprecedented morbidity, mortality, and disruption globally [1,2]. Despite severe complications that may be caused by the virus, a significantly high number of individuals did not require intensive care and continue to present sequelae months after the COVID-19 diagnosis [3].

Although COVID-19 is primarily manifested as a respiratory disease, many studies have shown that it is a systemic infection involving multiple systems and affecting individuals of all ages [2,4]. Initial studies showed that the harmful effects of COVID-19 could also reach the autonomic nervous system (ANS), even in non-hospitalized young individuals. Interestingly, detection of SARS-CoV-2 in carotid body of severe COVID-19 patients was previously reported, and its involvement may explain silent hypoxemia, a route of nervous system invasion and ANS deregulation [5,6,7]. It is hypothesized that autonomic dysfunction after COVID-19 infection could be mediated by the virus itself [8], or even associated with the cytokine storm response, resulting in oxidative stress. These mediators can subsequently cross the blood–brain barrier, ultimately increasing the activation of the sympathetic nervous system (SNS) [6,9]. Furthermore, autonomic dysfunction has also been shown to be associated with COVID-19 severity and prognosis [4].

Further investigations to clarify the impact of COVID-19 on ANS are needed as the ANS plays a major role in regulating the body’s homeostasis. Any dysfunction in this system can have detrimental effects on several physiological systems, including alterations in the cardiovascular system and the immune, hematological, and renal systems [10]. Additionally, a sympathetic overdrive has been identified as an independent predictor of mortality in several diseases [11,12,13]. Koopman et al., in an epidemiological study, observed that low cardiac autonomic modulation was associated with higher chances of mortality in individuals with low cardiovascular risk [14]. 

The autonomic imbalance may persist after acute SARS-CoV-2 infection, and the burden of several post-COVID-19 conditions, including postural orthostatic tachycardia syndrome (POTS), dysautonomia, fatigue, cerebral hypoperfusion syndrome, and impaired endothelial function, are associated with this condition independently of hospitalization [6,15,16]. A recent clinical case study reported strong benefits of invasive vagus nerve stimulation, such as relief of chest tightness and shortness of breath, in COVID-19 patients [17]. In fact, patients presenting post-viral condition, such as myalgic encephalomyelitis, induced by several viruses, manifest several alterations in ANS with harmful impact in the health and body homeostasis [18,19,20]. Thus, the ANS dysfunction after acute SARS-CoV-2 may be not necessarily new, but an old sequela observed in several post-viral infectious episodes. 

It is well established that other factors can directly affect autonomic function, such as obesity [21,22] and levels of physical activity [23,24]. Recent investigations indicate that obesity is associated with poor prognosis for SARS-CoV-2 infection, intensive care hospitalization, and disease progression [25,26]. Additionally, each 1-unit increase in body mass index (BMI) might be related to a 12% increase in the risk of severe COVID-19. These alterations can be associated with chronic low-grade inflammation and impairments in immune cell activation of obese patients [25,27]. However, it is necessary to investigate if overweight and obesity can cause additional changes to the ANS after COVID-19 infection. 

On the other hand, according to the World Health Organization, physical activity offers significant health benefits and mitigates health risks [28,29]. As a result, adults aged between 18 to 64 years should undertake at least 150 min of moderate- or 75 min of vigorous-intensity physical activity, or an equivalent combination of them per week to prevent detrimental effects on health outcomes [28]. The role of physical activity is still under investigation after COVID-19 infection, although sufficient physical activity decreases the prevalence of COVID-19-related hospitalization [30]. Therefore, it is important to understand if physical activity can play a protective role on the ANS after SARS-CoV-2 infection. Antagonic effects of BMI and physical activity were observed in the management of neurocognitive and fatigue symptoms, and in the incidence of post-acute sequelae SARS-CoV-2 infection (PASC) in healthcare workers [31]. 

This information is extremely relevant to help health professionals screen patients with COVID-19 presenting additional risk factors and educate and inform the global population of the factors that can influence the clinical course of this widespread disease. Persistent symptoms and physiological alterations associated with PACS impact physical and cognitive function and health-related quality of life, and there is a need to clarify the pathophysiological mechanisms of post-COVID-19 syndrome in young individuals previously infected by SARS-CoV-2 that do not require acute hospitalization.

The present study adds novelty and relevant aspects to the literature since it (1) investigates cases after mild to moderate COVID-19 infection in young adults, a subgroup of patients who are often overlooked, that did not require intensive care, addressing the influence of obesity and physical activity after COVID-19 infection on the ANS (2) through the estimation of the heart rate variability (HRV), a reliable and noninvasive method [32]. To our knowledge, these data are not yet available in the literature.

Thus, the primary aim of the study was to analyze the autonomic function of young adults after mild to moderate infection by COVID-19. Second, we aimed to identify the influence of body mass index (BMI) and levels of physical activity on the autonomic function in these individuals, indexed to the HRV indicators.

## 2. Materials and Methods

### 2.1. Ethical Approval

The study followed the ethical standards described by the Declaration of Helsinki. Additionally, all participants were notified about the study purpose and protocols and gave written informed consent to the protocols. The Ethical Institutional Review Board approved the study (approval number: 38701820.0.0000.5402).

### 2.2. Study Design

This is a cross-sectional observational study that is part of a broader research project: FIT-COVID Study [33]. The study was previously registered at the Brazilian Clinical Trials Registry (registration number: RBR-5dqvkv3). All reports followed the Strengthening the Reporting of Observational Studies in Epidemiology (STROBE) guidelines [34].

Patients that were infected with COVID-19 were recruited to participate in the study through local media (TV, radio) and social media and via electronic access to participants’ database of the Municipal Health Secretariat of Presidente Prudente São Paulo, Brazil (231,953 inhabitants; human development index, 0.806; 23,657 confirmed COVID-19 cases, moving average of 135 cases (May 2021)). We included male and female patients, aged 20–40 years after mild or moderate clinical COVID-19 with a previous positive PCR test and infection including slight clinical symptoms, fever, or respiratory symptoms, and that were not admitted to the intensive care unit. Participants were recruited after a minimum of 15 and maximum of 180 days of diagnosis by positive PCR test [35]. An age-matched healthy control group that was negative for COVID-19 was also recruited. To screen for confirmed or probable previous SARS-CoV-2 infection for the control group, a lateral flow test for IgM and IgG antibodies was conducted using internal anti-SARS-CoV-2 Immunoglobulin G (IgG) and Immunoglobulin M (IgM) ELISA kits.

The exclusion criteria were as follows: (1) presence of any chronic noncommunicable diseases, (2) smokers, (3) history of drug use, (4) medications such as anti-inflammatory drugs, antibiotics, and others known for their impact over the ANS, individuals that were vaccinated for COVID-19 and (5) patients that received intensive care.

### 2.3. Experimental Measures

#### 2.3.1. Initial Evaluation

A general anamnesis was taken, including sociodemographic characteristics and self-rated health and medical history (family comorbidities, cardiopulmonary symptoms, and medication use). Symptoms that emerged during acute COVID-19 infection and persistent symptoms were also evaluated. Participants were also assessed regarding their BMI and physical activity level. Finally, blood pressure tests after 20 min of rest were also performed using auscultatory blood pressure measurement [36]. 

#### 2.3.2. Body Mass Index

BMI was calculated according to previous literature. We considered BMI as the ratio of weight (kg) to height (m) squared. To evaluate the weight of participants, an electronic and calibrated scale (Kratos-Cas, São Paulo, SP, Brazil) was used with individuals wearing light clothes and barefoot. Height was measured using a portable anthropometer (Kratos-Cas, São Paulo, SP, Brazil) [37].

#### 2.3.3. Physical Activity Level

The level of physical activity was measured using a triaxial accelerometer (GT3X+; ActiGraph, LLC, Pensacola, FL, USA). Participants were instructed to wear the accelerometer above the waist for seven consecutive days during waking hours. A minimum of four days with at least 10 h/day was defined as valid accelerometer data. Participants were instructed not to use the accelerometer while bathing, sleeping, or performing water activities. Moreover, every morning, a researcher sent a WhatsApp message reminding the participant to use the accelerometer.

Non-wear periods were defined as time intervals of at least 60 consecutive minutes of zero counts, with an activity interruption allowance of 0–100 counts/min lasting a maximum of 2 consecutive minutes [38]. Counts per minute were calculated using the sum of the total activity counts in the vertical axis divided by the valid days. Sedentary time was defined as values <100 counts/min, light physical activity as values between 100 and 2019 counts/min, and moderate–vigorous physical activity as values >2020 counts/min. Data were processed using the ActLife software (version 6.9.2, Pensacola, FL, USA) [39].

#### 2.3.4. Heart Rate Variability

ANS function was measured through HRV, which is considered a simple, dependable, inexpensive, and noninvasive method [32]. For this evaluation, participants were asked to attend an outpatient clinic in a fasted state, having abstained from exercise, caffeine, chocolate, and alcohol for at least 24 h before evaluation and ≥4 h after a snack or light meal. Evaluations were performed in a silent space, with temperature of ∼23 °C. HRV analysis was undertaken in the morning to avoid circadian changes [32].

Heart rate was recorded beat-to-beat to evaluate cardiac autonomic modulation. We used a cardio-frequency meter (Polar RS800CX, Polar Electro, Kempele, Finland) at a 1 kHz sampling rate. Participants were sited with a chest strap and monitor and remained at rest with spontaneous breathing for 25 min. HRV was performed on 256 consecutive intervals between successive heartbeats (RR intervals) from the most stable segment on the tachogram. Only series with <5% error were considered suitable for analysis. Kubios HRV^®^ software (Biosignal Analysis and Medical Image Group, Department of Physics, University of Kuopio, Finland) was used to complete the HRV analysis [40,41].

The HRV was assessed in both the time and frequency domains. For the time domain, the mean RR intervals (reflecting global variability) were used. Additionally, the following indexes were calculated:

The square root of the mean squared difference between adjacent RR intervals (RMSSD) indicating parasympathetic modulations of the HR.

The standard deviation of all normal RR intervals (SDNN) reflecting global variability.

Geometric indexes from Poincaré plot SD1 and SD2, indicating parasympathetic and global modulations of HR, respectively.

Triangular interpolation of normal to normal RR intervals (TINN) indicating global variability.

Triangular index (RRtri), also signifying global variability.

For the frequency domain, spectral analysis was computed using the fast Fourier transform, and the following indexes were included in the study:

The low-frequency (LF: 0.04 Hz to 0.15 Hz) and high-frequency (HF: 0.15 Hz to 0.4 Hz) components were computed, representing sympathetic and parasympathetic activity, respectively. The LF to HF (LF/HF) ratio was calculated as a measure of autonomic (sympathovagal) balance.

The following indexes computed in Kubios HRV software were also included:

Baevsky’s stress index, a geometric measure of HRV reflecting cardiovascular system stress.

Parasympathetic nervous system index (PNS index) calculated based on the mean RR, RMSSD, and SD1 in normalized units.

SNS index, calculated based on the mean HR, Baevsky’s stress index, and SD2 in normalized units [41].

### 2.4. Statistical Analysis

Statistical procedures were performed using GraphPad Prism (version 5.00; GraphPad Software, San Diego, CA, USA) for Windows. Data distribution was analyzed using the Shapiro–Wilk test. For the primary analysis of comparisons between the post-COVID-19 and control groups, the unpaired *t*-test or Mann–Whitney U-test were used according to data distribution. The Fisher test was performed for categorical variables (sex).

In our secondary analysis, the post-COVID-19 and control groups were categorized according to the BMI, dividing each group into eutrophic (<25 kg/m^2^) and overweight/obese (≥25.0 kg/m^2^). Moreover, the groups were analyzed according to levels of physical activity, considering the subjects with <150 min of moderate to vigorous physical activity (MVPA) and subjects with ≥150 min of MVPA in each group [28]. All secondary analyses were performed using the one-way analysis of variance or Kruskal–Wallis test according to data distribution. Effect sizes (ESs) were calculated for significant secondary comparisons and classified as negligible (<0.01), small (0.1–0.29), medium (0.3–0.49), and large (>0.5) [42]. 

Additionally, binary logistic regression analysis was employed to identify factors associated with presence of post-COVID-19 and HRV indexes. Univariate and multivariate logistic regression were performed to test the association between the dependent and the independent variables.

Correlational analysis between the number of reported symptoms in the post-COVID-19 group and the HRV index was performed using Pearson or Spearman test according to data distribution. Independent variables that presented statistical correlation were inserted in a model of linear regression analysis. *p*-values < 0.05 were considered significant.

## 3. Results

For this study, 57 subjects were evaluated. After 19 exclusions due to errors in HRV recordings, 38 subjects with complete data were included in the final analysis. Figure 1 shows the flow diagram of the number of individuals at each stage of the study. In the post-COVID-19 group, nine subjects were women, and 11 were men. In the control group, 13 subjects were men, and 5 were women.

Patients in the post-COVID-19 group were evaluated 48.45 ± 31.41 days after testing positive for COVID-19. These individuals were classified as mild (*n* = 15; 75%) and moderate (*n* = 5; 25%) cases [43]. A mean of 5.75 ± 2.61 symptoms (clinical and respiratory) was noted. The main symptoms reported, and their respective prevalence, were respiratory symptoms (*n* = 15; 75%), headache (*n* = 14; 70%), body pain (*n* = 14; 70%), anosmia (*n* = 10; 50%), ageusia (*n* = 9; 45%), fever (*n* = 6; 30%), and diarrhea (*n* = 5; 25%). 

No differences in sex distribution were observed between the groups (*p* = 0.3276). As indicated in Table 1, the groups were uniformly distributed (*p* > 0.05) in terms of their descriptive characteristic data, except for light physical (*p* = 0.0444) and vigorous activities (*p* = 0.0380).

Table 2 and Figure 2 illustrate the primary analysis of the study, showing intergroup comparisons between HRV indexes in the post-COVID-19 and control groups. For the SNS activity, the post-COVID-19 group showed significantly increased levels of the stress index (*p* = 0.0273).

Analysis of the parasympathetic activity showed that the post-COVID-19 group presented significant lower values of RMSSD (*p* = 0.0452) and SD1 (*p* = 0.0118). Additionally, participants in the post-COVID-19 group also presented lower global variability, reflected through the SDNN (*p* = 0.0282), TINN (*p* = 0.0404), and SD2 (*p* = 0.0145) indexes.

In an univariable binary logistic regression, the influence of post-COVID-19 on HRV was analyzed (Appendix A). We observed that the presence of post-COVID0.-19 was significantly associated (*p* < 0.05) with sympathetic activity (stress index: OR: 1.22; CI 95%: 1.00 to 1.48; *p* = 0.041), parasympathetic activity (RMSSD: OR: 0.89; CI 95%: 0.81 to 0.98; *p* = 0.026 and SD1: OR:0.85; CI 95%: 0.74 to 0.98; *p* = 0.027) and global variability (SDNN: OR: 0.92; CI 95%: 0.85 to 0.99; *p* = 0.039 and SD2: OR: 0.93; CI 95%: 0.87 to 0.99; *p* = 0.024). This data is presented in Appendix A.

In the secondary analysis, the post-COVID-19 and control groups were categorized into non-obese and overweight/obese, as shown in Table 3 and Figure 3. Overweight/obese (Ow/Ob) patients in the post-COVID-19 group presented significantly lower parasympathetic activity in the SD1 index (95% confidence interval (CI) = −15.33 to −0.3959; *p* = 0.0301; ES post-COVID-19 Ow/Ob vs. post-COVID-19 Non-obese = 0.79; ES post-COVID-19 Ow/Ob vs. control Ow/Ob = 0.92).

Moreover, these subjects presented with reduced global variability, as shown by SDNN (95% CI = −23.29 to −1.453; *p* = 0.024; ES post-COVID-19 Ow/Ob vs. post-COVID-19 Non-obese = 0.94; ES post-COVID-19 Ow/Ob vs. control Ow/Ob = 0.77), TINN (95% CI = −90.41 to −2.826; *p* = 0.0288; ES post-COVID-19 Ow/Ob vs. COVID-19 Eutro = 0.96; ES post-COVID-19 Ow/Ob vs. control Ow/Ob = 0.94), and SD2 (95% CI = −29.45 to −1.899; *p* = 0.0179; ES post-COVID-19 Ow/Ob vs. post-COVID-19 Non-obese = 0.92; ES post-COVID-19 Ow/Ob vs. control Ow/Ob = 1.03) indexes when compared with those in non-obese individuals in the control group.

Comparisons between groups were also investigated with regard to the levels of physical activity, dividing the post-COVID-19 and control groups according to the MVPA (≥150 min and <150 min), as shown in Table 4 and Figure 4.

Physically inactive subjects (<150 min of MVPA) in the post-COVID-19 group presented with significantly higher levels of stress index compared to those in active subjects in the control group (0.0147; ES post-COVID-19 < 150 min vs. post-COVID-19 ≥ 150 min = 0.94; ES post-COVID-19 < 150 min vs. control < 150 min = 1.22). PNS activity was significantly increased in physically active subjects in the control group compared to that in the physically inactive post-COVID-19 group (<150 min of MVPA).

These results were demonstrated through the RMSSD (ES post-COVID-19 < 150 min vs. post-COVID-19 ≥ 150 min = 0.92; ES post-COVID-19 < 150 min vs. control < 150 min = 0.80), pNN50 (ES post-COVID-19 < 150 min vs. post-COVID-19 ≥ 150 min = 1.16; ES post-COVID-19 < 150 min vs. control < 150 min = 0.71), and SD1 (ES post-COVID-19 < 150 min vs. post-COVID-19 ≥ 150 min = 0.92; ES post-COVID-19 < 150 min vs. control < 150 min = 1.00) indexes (*p* < 0.05).

Global variability was significantly reduced in the physically inactive post-COVID-19 group (<150 min of MVPA) when compared with that in the physically active patients in the control group. These results can be observed in the SDNN (ES post-COVID-19 < 150 min vs. post-COVID-19 ≥150 min = 0.83; ES post-COVID-19 < 150 min vs. control < 150 min = 0.69), TINN (ES post-COVID-19 < 150 min vs. post-COVID-19 ≥ 150 min = 1.04; ES post-COVID-19 < 150 min vs. control < 150 min = 0.82), and SD2 (ES COVID-19 < 150 min vs. post-COVID-19 ≥ 150 min = 0.79; ES post-COVID-19 < 150 min vs. control < 150 min = 0.88) indexes (*p* < 0.05).

Additionally, the combination of overweight/obesity and low levels of physical activity was investigated to determine if it could impact the ANS in non-obese and physically active individuals, as shown in Table 5. Identified participants who were overweight/obese with an MVPA <150 min in the post-COVID-19 group presented significantly higher values of sympathetic activity (*p* = 0.0149) compared with those in eutrophic participants with an MVPA >150 min in the control group. Furthermore, parasympathetic activity and global variability are notably reduced in participants who were overweight/obese with an MVPA <150 min in the post-COVID-19 group.

Additionally, we performed a multivariable regression analysis, designed to test the possibility of confounding relations of HRV with other clinically relevant independent variables (BMI and physical activity levels). We identified that RMSSD (OR: 1.20; CI 95%: 0.98 to 1.48; *p* = 0.071), SD1 (OR: 0.86; CI 95%: 0.74 to 0.99; *p*= 0.045) and SD2 (OR: 0.93; CI 95%: 0.87 to 0.99; *p* = 0.046) still held a significant association with post COVID-19 presence when adjusted for BMI (Appendix B).

When MVPA were included in the multivariable regression analysis model, only SDNN (OR: 0.89; CI 95%: 0.80 to 1.00; *p* = 0.049 maintained significant association with post COVID-19 presence, and the other HRV indexes held a significant association with post-COVID-19 (*p* > 0.05), showing that MVPA might be a confounding outcome for this investigation.

## 4. Discussion

This study analyzed if COVID-19 may affect the autonomic function of young adults after mild and moderate infections. The relationship between BMI and physical activity level with autonomic function was also investigated. Our main finding is that even in mild and moderate infections, young adults who had COVID-19 had greater sympathetic activity, less parasympathetic activity, and global variability when compared to that in uninfected individuals. Moreover, in participants who were overweight and obese and/or physically inactive, cardiac autonomic modulation showed worse indices. Collectively, our study provides new insights regarding the role of BMI and physical activity status on post-COVID-19-infection autonomic deregulation that may contribute to the understand of pathophysiology and treatment of PACS.

Corroborating our findings, a recent study by Stute et al. [5] also showed that young adults recovering from COVID-19 presented with autonomic dysregulation. The study used an invasive method (muscle sympathetic nerve activity) in 16 subjects after COVID-19 infection, which contrasts with our study that showed similar results through HRV analysis, a simple, dependable, inexpensive, and noninvasive method [32]. Using wearable sensor data, Radin and colleagues [44] reported a prolonged physiological impact of SARS-CoV-2 infection, lasting approximately 2–3 months, on resting heart rate which may reflect ANS dysfunction. 

The prolonged alterations in autonomic function may be related to a persistent systemic inflammatory condition observed in post-viral stage. These findings may be attributed to the increased state of inflammation generated during COVID-19 infection, as well as the direct infusion of inflammatory cytokines [8,9]. These mechanisms are often characterized by marked increases in SNS activity [5,45]. The presence of markers of oxidative stress in COVID-19 and other viral diseases has been observed [8,46]. Therefore, we believe that the oxidative stress and subsequent release of inflammatory cytokines that accompany SARS-CoV-2 infection could explain the alterations in the ANS in young subjects after COVID-19 in our study. Other factors explaining ANS alterations may also be attributed to behavioral changes during and after infection, such as changes in physical activity [47], nutritional status [48], and food and fluid consumption [49]. These conditions also affect the blood volume (causing hypovolemia) and cardiovascular response [5].

This result highlights the urgency of assessments regarding the progression of COVID-19 in patients who have recovered, even if they are young or had mild or moderate symptoms, as an ANS dysregulation reveals a body homeostasis dysfunction and can predict cardiovascular and metabolic illnesses and is directly correlated with mortality in several diseases [10,50].

It is known that COVID-19 can affect different organs and systems at different magnitudes [6,45]. However, the impact of COVID-19 on the autonomic function of young individuals and its relationship with the BMI and physical activity level are not yet well established. Our findings revealed that patients who were overweight/obese in the post-COVID-19 group presented significantly lower parasympathetic activity and reduced global variability compared to that in eutrophic individuals in the control group (Table 3 and Figure 3). Autonomic dysfunction after COVID-19 can be aggravated in individuals with obesity, as studies previously showed that reduced HRV is associated with obesity and inflammation [51]. The chronic low-grade inflammation caused by obesity [52], combined with the cytokine storm response from the COVID-19 infection [9], may promote, and even worsen, autonomic dysregulation in overweight/obese patients.

Our study showed that in physically active participants, cardiac autonomic modulation levels were better when compared to those in their inactive peers, even in participants who had COVID-19. However, multivariable regression analysis showed vigorous physical activity as a possible confounding factor, showing that the differences observed in HRV might be primarily modulated by levels of physical activity and not by the presence of post-COVID-19. It is known that exercise promotes positive adaptations in HRV parameters in both control subjects and patients suffering from a variety of diseases [23,50]. Similarly, both exercise training and physical activity habits improve heart rate variability parameters, the autonomic profile, and arterial compliance, as well as the baroreflex sensitivity in HIV+ patients, highlighting the impact of lifestyle habits to modulate ANS in viral conditions [53,54].

Regular exercise can promote functional and structural changes in the central and peripheral mechanisms of the cardiovascular system to ensure proper blood perfusion and cardiac response according to metabolic demands [55]. There is a consistent body of evidence stating that exercise increases resting vagal activity and induces neuronal plasticity in central autonomic networks. It is suggested that these mechanisms are mainly related to improved control of the nucleus tractus solitarii, rostral ventrolateral medulla, and paraventricular nucleus of the hypothalamus [55,56]. The previous autonomic adaptations in physically active individuals may explain the protective role of exercise observed after COVID-19 infection.

Post-viral conditions, such as chronic fatigue syndrome, are characterized by autonomic dysfunction that strongly interfere in the health quality and neurocognitive aspects of the patients. Physical activity is a consistent predictor of the measure of autonomic function in chronic fatigue syndrome patients, predicting 45% of the variance in the resting HR, 36% of the variance in the sitting HR, 20% of the variance in standing HR, 20% of the variance in HR on tilt, and 56% of the variance in the SBP fall in phase II of the Valsalva maneuver [20]. 

On the other hand, associations between low levels of moderate- to high-intensity physical activity patterns with resting HR and reduced HRV were previously reported in chronic fatigue syndrome patients, suggesting that reduction in physical activity can, in part, be explained by autonomic dysfunction but not fatigue severity [57]. Here, we report that post COVID-19 patients had significantly low levels of vigorous physical activity that may be related to the imbalance in ANS dysfunction. 

Additionally, this study revealed that the number of symptoms reported by patients during the course of the COVID-19 infection was associated with the HRV response. These data may be explained by the severity of the virus infection that will usually promote more severe symptoms, consequently affecting levels of inflammation and oxidative stress [3,8], mediating changes in the HRV [58].

The main limitations to our investigation were (1) the cross-sectional nature of the study and the small sample size (*n* = 57). This is of particular concern because our primary outcome measure (HRV) is characterized by high interindividual variability. (2) Despite matching for age and BMI, groups were not similar regarding physical activity. The post-COVID-19 group was less active, probably due to the symptoms and indisposition related to the infection, which can generate a confounding bias. Therefore, longitudinal designs monitoring healthy young subjects recovering from SARS-CoV-2 infection are undoubtedly warranted to better understand the impact of the virus on ANS activity.

This study also presents several strengths. First, young subjects with post mild and moderate COVID-19 infection were investigated, a commonly overlooked population in recent studies. Second, to our knowledge, this is the first study to investigate the influence of the BMI and physical activity level on the ANS after COVID-19. Third, our results are of great clinical relevance as they show the importance of screening and monitoring in young adults, especially those who are overweight/obese or physically inactive. Finally, in our study, the practice of physical activity was measured using accelerometry, which provides more reliable information about the intensity of physical activity, as self-reports may be susceptible to memory bias.

As practical applications of the present study, the importance of maintaining the BMI close to eutrophic levels and the consistent practice of physical activity is highlighted, even in the time of a pandemic where social isolation may be necessary (considering all precautions, for example, when performing outdoor physical activities).

## 5. Conclusions

Mild to moderate SARS-CoV-2 infection promotes changes in the ANS of young adults that persist one month after the acute phase of the disease. Additionally, the BMI can modulate the changes in the ANS after COVID-19 in different perspectives: excessive BMI accentuates the changes in the ANS. Finally, the severity of the symptomatology maintained after the SARS-CoV-2 infection may also have important consequences on the daily life activities and ability to perform physical activities, thus additionally contributing to the autonomic balance of these individuals.

## Figures and Tables

**Figure 1 ijerph-19-02457-f001:**
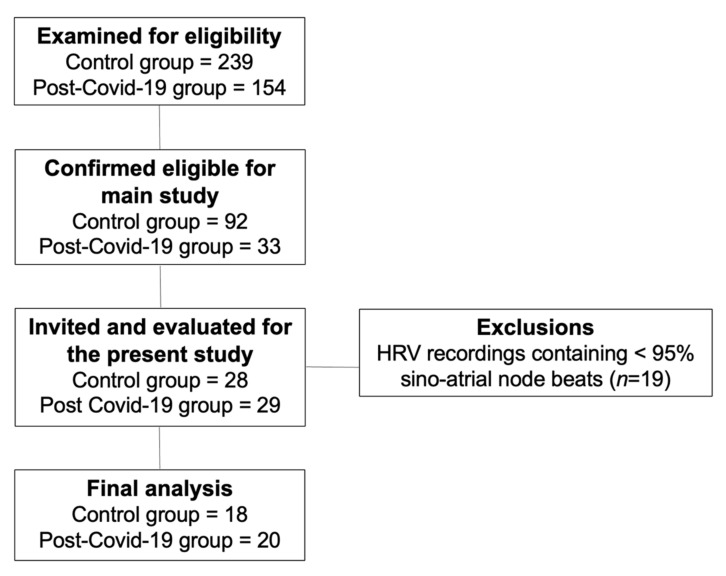
Flow diagram of the study.

**Figure 2 ijerph-19-02457-f002:**
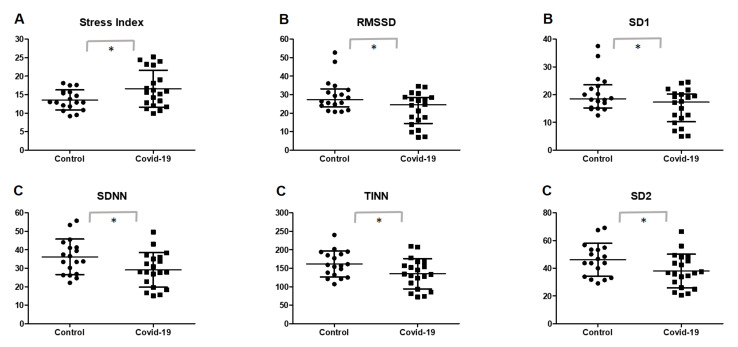
Scatterplot of HRV index comparison between the post-COVID-19 and control groups. Data are expressed as mean and standard deviation for data with normal distribution and median and 25–75% interquartile range for data with non-normal distribution. (**A**) Sympathetic nervous system activity; (**B**) parasympathetic nervous system activity; (**C**) global variability. * Statistical significance (*p* < 0.05) between the post-COVID-19 and control groups; RMSSD, root mean square of differences between adjacent normal RR intervals in a time interval; SD1, standard deviation of instant beat-to-beat variability; SDNN, standard deviation of all normal RR intervals recorded in a time interval; TINN, triangular interpolation of NN interval; SD2, standard deviation of the long-term interval between consecutive heartbeats.

**Figure 3 ijerph-19-02457-f003:**
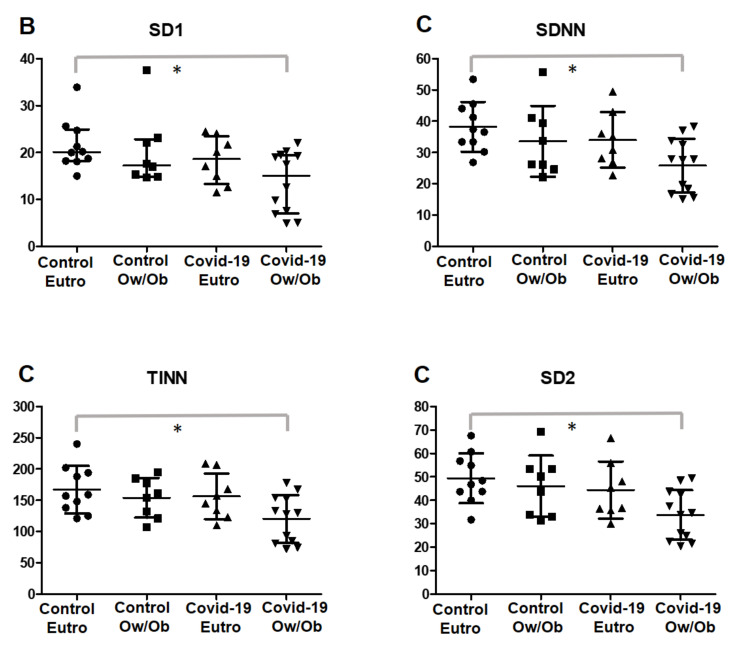
Scatterplot of HRV indexes according to the BMI data with normal distribution are expressed as mean and standard deviation and data with non-normal distribution as median and 25–75% interquartile range. (**B**) parasympathetic nervous system activity; (**C**) global variability. * Statistical differences between groups; SD1, standard deviation of instantaneous beat-to-beat variability; SDNN, standard deviation of all normal RR intervals recorded in a time interval; TINN, triangular interpolation of NN interval; SD2, standard deviation of long-term intervals between consecutive heartbeats.

**Figure 4 ijerph-19-02457-f004:**
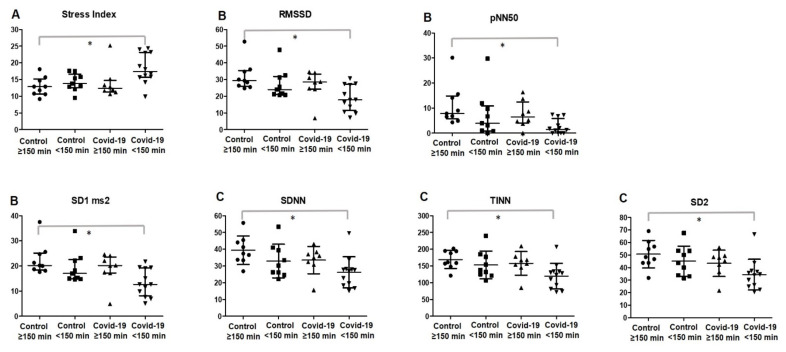
Scatterplot of HRV indexes according to the physical activity levels. Data with normal distribution are expressed as mean and standard deviation and data with non-normal distribution as median and 25–75% interquartile range. (**A**) Sympathetic nervous system activity; (**B**) parasympathetic nervous system activity; (**C**) global variability. * Statistical significance (*p* < 0.05) between the control group with ≥150 min of MVPA and the COVID-19 group with <150 min of MVPA. RMSSD, root mean square of differences between adjacent normal RR intervals in a time interval; pNN50, percentage of adjacent RR intervals with a difference of duration >50 ms; SD1, standard deviation of instantaneous beat-to-beat variability; SD2, standard deviation of long-term intervals between consecutive heartbeats; SDNN, standard deviation of all normal RR intervals recorded in a time interval; TINN, triangular interpolation of NN interval.

**Table 1 ijerph-19-02457-t001:** Sample characterization between groups. Data with normal distribution are expressed as mean and standard deviation and data with non-normal distribution as median and 25–75% interquartile range.

Groups	Control		Post-COVID-19				
Variables	Mean/ Median	SD/IQR	Mean/ Median	SD/IQR	Diff between Means	95%CI	*p*-Value
Sex (F/M)	(5/13)	-	(9/11)	-	-	-	0.3276
Age (years)	26.22	5.22	29.17	6.32	−2.95 ± 1.89	−0.89 to 6.80	0.1276
Weight (kg)	74.59	17.32	76.57	14.36	−1.98 ± 5.14	−8.45 to 12.42	0.7023
Height (m)	1.69	0.09	1.73	0.08	−0.03 ± 0.02	−0.09 to 0.01	0.1712
BMI (kg/m^2^)	24.67	4.79	26.76	5.12	−2.09 ± 1.61	−1.18 to 5.36	0.2039
Systolic blood pressure	120.00	120–120	120.00	120–130	8.55 ± 5.02	−1.64 to 18.75	0.1189
Diastolic blood pressure	80.00	80–80	80.00	80–80	0.05 ± 1.06	−2.11 to 2.22	0.9588
Sedentary activity (min/day)	578.40	534.90–676.70	499.60	467.10–593.50	55.99 ± 97.31	−254.70 to 142.70	0.5693
Light activity (min/day)	239.00	65.15	309.00	116.5	−70.01 ± 33.37	1.87 to 138.10	0.0444 *
Moderate activity (min/day)	17.90	5.10–29.90	13.36	9.25–33.72	0.03 ± 8.05	−16.48 to 16.41	0.9966
Vigorous activity (min/day)	3.64	0.0–15.80	0.00	0.00–0.50	−7.931 ± 3.04	−14.14 to −1.72	0.0380 *
MVPA (min/day)	23.72	12.60–46.00	14.79	9.57–34.04	8.58 ± 9.94	−28.89 to 11.72	0.3964

SD, standard deviation; IQR, interquartile range, F, female, M, male; kg, kilogram; m^2^, square meter; min, minutes; BMI, body mass index; MVPA, moderate to vigorous physical activities. * Statistical significance (*p* < 0.05) between the post-COVID-19 and control groups

**Table 2 ijerph-19-02457-t002:** Comparisons between HRV indexes in the post-COVID-19 and control groups. Data with normal distribution are expressed as mean and standard deviation and data with non-normal distribution as median and 25–75% interquartile range.

Groups	Control		Post-COVID-19				
Variables	Mean/ Median	SD/[IQR]	Mean/Median	SD/[IQR]	Diff between Means	95%CI	*p*-Value
Sympathetic nervous system activity							
Mean HR	79.83	8.85	83.85	13.38	4.01 ± 3.72	−3.54 to 11.58	0.2882
Stress index	13.57	2.72	16.62	4.98	3.04 ± 1.32	0.35 to 5.72	0.0273 *
LF (nu)	65.49	15.30	66.61	15.65	1.12 ± 5.03	−9.09 to 11.33	0.825
SNS index	1.59	0.89	2.37	1.57	0.77 ± 0.44	−0.07 to 1.63	0.0719
Parasympathetic nervous system activity							
Mean RR	738.50	694.0–846.50	689.50	648.0–800.0	−25.94 ± 37.00	−101.0 to 49.15	0.1834
RMSSD	27.40	23.40–33.15	24.45	14.40–28.55	−7.69 ± 2.88	−13.55 to −1.83	0.0452 *
HF (nu)	34.49	15.30	33.33	15.65	−1.15 ± 5.03	−11.36 to 9.05	0.8198
pNN50	6.73	3.74–12.50	3.41	0.25–7.36	−4.70 ± 2.23	−9.24 to 0.17	0.055
SD1 (ms)	19.35	16.58–23.50	17.30	10.23–20.25	−5.42 ± 2.04	−9.58 to −1.27	0.0118 *
PNS index	−1.21	−1.49–(−0.90)	−1.60	−2.13–(−0.94)	−0.36 ± 0.23	−0.83 to 0.11	0.0845
Global variability							
SDNN	36.17	9.59	29.13	9.37	−7.04 ± 3.07	−13.29 to −0.79	0.0282 *
RR triangular index	9.02	1.91	7.91	2.34	−1.10 ± 0.69	−2.52 to 0.31	0.1218
TINN	161.30	34.93	135	40.77	−26.33 ± 12.39	−51.47 to −1.19	0.0404 *
LF/HF	1.99	1.47–3.39	1.83	1.11–3.88	0.57 ± 0.71	−0.86 to 2.01	0.9883
SD2 (ms)	47.93	11.53	38.02	12.17	−9.91 ± 3.85	−17.74 to −2.08	0.0145 *

HR, heart rate; LF, low-frequency component; ms, millisecond; nu, normalized unit; SNS index, sympathetic nervous system index; RMSSD, root mean square of differences between adjacent normal RR intervals in a time interval; HF, high-frequency component; pNN50, percentage of adjacent RR intervals with a difference in duration >50 ms; SD1, standard deviation of instantaneous beat-to-beat variability; SD2, standard deviation of long-term intervals between consecutive. heartbeats; PNS index, parasympathetic nervous system index; SDNN, standard deviation of all normal RR intervals recorded in a time interval; TINN, triangular interpolation of NN interval. * Statistical significance (*p* < 0.05) between post-COVID-19 and control group.

**Table 3 ijerph-19-02457-t003:** Comparison of HRV indexes between the post-COVID-19 and control groups according to the BMI. Data with normal distribution are expressed as mean and standard deviation and data with non-normal distribution as median and 25–75% interquartile range.

	Control				Post-COVID-19				
Groups	Non-Obese (*n* = 10)		Overweight/Obese (*n* = 8)		Non-Obese (*n* = 8)		Overweight/Obese (*n* = 12)		
Variables	Mean/Median	SD/[IQR]	Mean/ Median	SD/[IQR]	Mean/ Median	SD/[IQR]	Mean/ Median	SD/[IQR]	*p*-Value
Sympathetic activity									
Mean HR	83.10	8.43	75.75	8.03	84.63	14.01	83.33	13.55	0.2421
Stress index	13.70	3.01	13.41	2.49	14.59	3.20	17.97	5.59	0.2276
LF (nu)	61.95	17.65	69.92	11.31	58.42	10.90	72.08	16.31	0.1643
SNS index	1.82	0.94	1.29	0.76	2.09	1.32	2.55	1.74	0.2184
Parasympathetic activity									
Mean RR	728.80	79.83	800	90.11	727.50	141.30	739.2	131.4	0.2339
RMSSD	28.90	(25.58–35.13)	24.40	(21.00–32.25)	26.30	(18.73–33.23)	21.25	(9.95–27.53)	0.0786
HF (nu)	38.03	17.65	30.06	11.3	41.54	10.91	27.87	16.3	0.1142
pNN50	7.32	(6.00–14.49)	4.45	(0.83–11.40)	4.96	(2.04–12.16)	1.54	(0–7.07)	0.0506
SD1 (ms)	20.10	(18.18–24.93)	17.3	(14.93–22.85)	18.60	(13.28–23.50)	15.05	(7.075–19.45)	0.0301 *
PNS index	−1.33	(−1.53–(−0.92))	−0.97	(−1.45–(−0.82))	−1.48	(−2.08–(−0.95))	−1.65	(−2.17–(−0.71))	0.3112
Global variability									
SDNN	38.22	7.95	33.61	11.34	34.05	8.87	25.85	8.50	0.024 *
RR triangular index	9.58	2.26	8.31	1.12	9.04	2.04	7.15	2.29	0.0507
TINN	167.20	37.96	154	31.63	156.60	36.57	120.60	38.11	0.0288 *
LF/HF	2.06	1.14	2.91	1.87	1.58	0.81	3.97	3.02	0.0605
SD2 (ms)	49.45	10.61	46.03	13.06	44.38	12.20	33.78	10.6	0.0179 *

HR, heart rate; LF, low-frequency component; ms, millisecond; nu, normalized unit; SNS index, sympathetic nervous system index; RMSSD, root mean square of differences between adjacent normal RR intervals in a time interval; HF, high-frequency component; pNN50, percentage of adjacent RR intervals with a difference in duration >50 ms; SD1, standard deviation of instantaneous beat-to-beat variability; SD2, standard deviation of long-term intervals between consecutive heartbeats; PNS index, parasympathetic nervous system index; SDNN, standard deviation of all normal RR intervals recorded in a time interval; TINN, triangular interpolation of NN interval. * Statistical differences between non-obese patients in the control group and overweight/obese patients in the post-COVID-19 group.

**Table 4 ijerph-19-02457-t004:** Comparisons of HRV indexes between the post-COVID-19 and control groups according to the physical activity levels. Data with normal distribution are expressed as mean and standard deviation and data with non-normal distribution as median and 25–75% interquartile range.

	Control Group				Post-COVID-19				
Groups	≥150 min MVPA (*n* = 9)		<150 min MVPA (*n* = 9)		≥150 min MVPA (*n* = 8)		<150 min MVPA (*n* = 12)		
Variables	Mean/ Median	SD/[IQR]	Mean/ Median	SD/[IQR]	Mean/Median	SD/[IQR]	Mean/Median	SD/[IQR]	*p*-Value
Sympathetic activity									
Mean HR	80.33	9.77	79.33	8.39	77.75	18.13	87.92	7.41	0.1755
Stress index	12.90	(10.70–15.20)	13.80	(12.50–16.65)	12.40	(11.33–14.73)	17.40	(15.68–23.15)	0.0147 *
LF (nu)	60.08	18.54	70.91	9.37	65.20	13.12	67.56	17.64	0.5058
SNS index	1.52	0.95	1.65	0.86	1.57	1.88	2.90	1.10	0.374
Parasympathetic activity									
Mean RR	757.40	104.10	763.40	78.92	806.50	178.10	686.5	59.63	0.1113
RMSSD	29.40	(25.95–35.45)	24	(21.00–31.90)	28.50	(24.33–33.33)	17.85	(11.48–27.15)	0.0148 *
HF (nu)	39.90	18.54	29.08	9.37	34.78	13.11	32.37	17.63	0.5044
pNN50	7.84	(5.76–14.85)	3.95	(0.89–10.85)	6.48	(4.04–12.46)	1.45	(0.0–5.82)	0.0126 *
SD1 (ms)	20.20	(18.40–25.15)	17	(14.90–22.60)	20.20	(17.20–23.58)	12.65	(8.15–19.23)	0.0155 *
PNS index	−1.32	(−1.45–(−0.87))	−1	(−1.62–(−0.86)	−0.87	(−1.79–(−0.13))	−1.82	(−2.17-–(−1.33))	0.317
Global variability									
SDNN	39.40	8.40	32.94	10.07	33.54	8.14	26.19	9.27	0.0203 *
RR triangular index	9.52	(8.13–10.46)	8.45	(6.94–8.97)	8.92	(8.10–10.23)	7.65	(4.97–8.40)	0.0509
TINN	169.40	26.93	153.20	41.47	157.90	35.25	119.80	38.02	0.0192 *
LF/HF	1.64	(1.06–3.30)	2.61	(1.64–3.43)	1.49	(1.20–3.45)	2.67	(0.95–4.23)	0.6431
SD2 (ms)	48.40	(43.80–58.75)	43.70	(33.45–53.35)	46.65	(37.90–49.15)	34.25	(25.18–37.25)	0.0184 *

HR, heart rate; LF, low-frequency component; ms, millisecond; nu, normalized unit; SNS index, sympathetic nervous system index; RMSSD, root mean square of differences between adjacent normal RR intervals in a time interval; HF, high-frequency component; pNN50, percentage of adjacent RR intervals with a difference of duration >50 ms; SD1, standard deviation of instantaneous beat-to-beat variability; SD2, standard deviation of long-term intervals between consecutive heartbeats; PNS index, parasympathetic nervous system index; SDNN, standard deviation of all normal RR intervals recorded in a time interval; TINN, triangular interpolation of NN interval. * Statistical significance (*p* < 0.05) between the control group with ≥150 min of MVPA and the post-COVID-19 group with <150 min of MVPA.

**Table 5 ijerph-19-02457-t005:** Comparison of HRV indexes between eutrophic and physically active subjects that presented with both overweight/obesity status and low levels of physical activity in both groups. Data are expressed as median and 25–75% interquartile range.

	Control				Post-COVID-19				
Groups	Non-obese + >150 min MVPA (*n* = 7)		Ow/Ob + <150 min MVPA (*n* = 6)		Non-obese + >150 min MVPA (*n* = 4)		Ow/Ob + <150 min MVPA (*n* = 8)		
Variables	Mean/ Median	SD/[IQR]	Mean/ Median	SD/[IQR]	Mean/ Median	SD/[IQR]	Mean/ Median	SD/[IQR]	*p*-Value
Sympathetic activity									
Mean HR	84	(75–87.00)	77.5	(70.75–81.00)	84.5	(63.50–94.25)	90	(82.50–91.00)	0.0772
Stress index	12.9	(10.80–15.70)	13.4	(12.70–16.25)	13.05	(11.98–14.73)	19.6	(15.68–23.80)	0.0149 *
LF (nu)	62.13	(41.57–72.84)	67.55	(60.43–80.16)	55.19	(51.51–58.73)	78.63	(48.76–86.92)	0.2434
SNS index	1.55	(0.92–2.52)	1.365	(1.14–1.92)	1.68	(0.41–2.71)	3.11	(2.05–4.00)	0.0237 **
Parasympathetic activity									
Mean RR	716	(688.0–795.0)	772.5	(744.0–847.3)	712	(636.0–958.3)	668.5	(658.0–727.0)	0.0659
RMSSD	29.4	(26.4–34.80)	22.8	(20.78–31.55)	31.25	(25.25–34.48)	15.8	9.95–27.15	0.0174 *
HF (nu)	37.89	(27.14–58.37)	32.44	(19.82–39.55)	44.79	(41.23–48.48)	21.21	(13.06–51.18)	0.2434
pNN50	7.84	(6.57–14.12)	2.475	(0.58–10.29)	9.47	(4.83–15.70)	0.5	(0.0–5.60)	0.0215 *
SD1 (ms)	20.2	(18.70–24.70)	16.15	(14.78–22.35)	22.1	(17.85–24.40)	11.2	(7.07–19.23)	0.0174 *
PNS index	−1.32	(−1.410–(−0.98))	−0.975	(−1.41–(−0.90))	−1.275	(−1.798–(−0.025))	−1.825	(−2.17–(−1.33))	0.5407
Global variability									
SDNN	37.4	(33.40–44.10)	26.2	(23.98–39.80)	35.5	(31.93–41.33)	23.6	(17.13–27.88)	0.0128 *
RR triangular index	9.52	(8.0–11.06)	8.09	(6.78–8.89)	8.92	(8.76–11.38)	6.81	(4.51–8.40)	0.0374 *
TINN	159	(148.0–194.0)	143	(117.5–179.0)	162.5	(148.0–198.8)	110.5	(75.75–132.3)	0.0123 *
LF/HF	1.641	(0.71–2.68)	2.15	(1.53–4.31)	1.23	(1.06–1.42)	3.703	(0.95–7.06)	0.2434
SD2 (ms)	48.4	(43.80–56.80)	42	(32.63–53.33)	46.65	(1.67–2.68)	29.95	(23.10)	0.0198 *

HR, heart rate; LF, low-frequency component; ms, millisecond; nu, normalized unit; SNS index, sympathetic nervous system index; RMSSD, root mean square of differences between adjacent normal RR intervals in a time interval; HF, high-frequency component; pNN50, percentage of adjacent RR intervals with a difference of duration >50 ms; SD1, standard deviation of instantaneous beat-to-beat variability; SD2, standard deviation of long-term intervals between consecutive heartbeats; PNS index, parasympathetic nervous system index; SDNN, standard deviation of all normal RR intervals recorded in a time interval; TINN, triangular interpolation of NN interval. * Statistical significance (*p* < 0.05) between non-obese patients with >150 min MVPA in the control group and overweight/obese patients with <150 min MVPA in the post-COVID-19 group. ** Statistical significance (*p* < 0.05) between overweight/obese patients with <150 min MVPA in the control group and overweight/obese patients with <150 min MVPA in the post-COVID-19 group.

## Data Availability

Availability of these data is open at https://data.mendeley.com/datasets/t45r8yd6jd/1 (accessed on 29 November 2021). Freire, Ana Paula (2021), “Body mass and physical activity in autonomic function modulation on post-COVID-19 condition: an observational subanalysis of FIT-COVID study”, Mendeley Data, V1, doi:10.17632/t45r8yd6jd.1, [59].

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
