# Peer review of "Role of Body Mass and Physical Activity in Autonomic Function Modulation on Post-COVID-19 Condition: An Observational Subanalysis of Fit-COVID Study"

_ijerph, 2022, doi:10.3390/ijerph19042457_

Round 1

Reviewer 1 Report

The aim of the paper is to assess ANS and identify the influence of BMI and physical activity levels on autonomic function in young adults after mild to moderate COVID-19 infection.

The topic is interesting and topical. However, the study has shortcomings that prevent it from being published. 

  1. The number of subjects (57) is very small for a cross-sectional study. Usually hundreds of patients are evaluated in this type of study. The division into groups was designed in such a way that an assessed group would consist of only few individuals.
  2. The presentation of the results is very difficult to read. It should be simplified with references to tables 
  3. Methods include whole paragraphs plagiarised (see file)
  4. There are non-significant but important discrepancies between the two study groups (e.g. BMI), which could influence the differences found for ANS.
  5. Rather than the influence of COVID, it would seem that it is the degree of physical activity and BMI that influence the ANS.
  6. If the authors want to use such a small sample I think it is necessary to revise the division into groups, adjusting for BMI. However, we should remember that cross-sectional studies do not allow to associate a risk factor with a disease with any certainty, since they do not allow to verify with certainty which risk factor a 'sick person' has come into contact with. This should be highlighted in the discussion.

Author Response

To: Stefan Krstovic

Assistant Editor, MDPI Belgrade - IJERPH Editorial Office

December 20th, 2021

Thank you for the opportunity to have our paper reviewed and for the chance to resubmit it. We found the reviewers’ comments very appropriate to improve our manuscript and would like to thank them for their valuable contribution.

Please, find bellow a point-by-point reply to the reviewers’ comments.

We hope that this updated version will reach the standards of your esteemed journal and look forward hearing from you.

Yours sincerely,

Dr. Ana Paula Coelho Figueira Freire

Assistant Professor

Central Washington University

Department of Health Sciences

Ellensburg, WA, 98926 - USA

Reviewer 2 Report

Introduction - In addition to the information presented, the authors should also present the WHO recommendations on physical activity for the age group used in this study. Also, the citation system used in the article is not in accordance with the citation system imposed by the journal, please read the instructions for the authors.

There is too little information / study presented in the Introduction about PA and body mass, there are two very important variables in this study, please expand the chapter with new information on this level.

Can you send us the proof of the ethics committee's agreement? I do not understand why this agreement consists so many numbers, i.e. ''38701820.0.0000.5402''

What happened or how the GT3X was used (ActiGraph, LLC, Pensacola, FL, USA) when the participants sleep? I do not find this information in the methodology, only when they took a bath, please fill in the necessary information

References: please put in accordance with journal policy

Author Response

  1. 1. Introduction - In addition to the information presented, the authors should also present the WHO recommendations on physical activity for the age group used in this study. Also, the citation system used in the article is not in accordance with the citation system imposed by the journal, please read the instructions for the authors.

Answer: We agree and thank you for your comment. The WHO recommendations on physical activity were included in the introduction (see below). Additionally, the citation system was corrected according instruction of the journal. All modifications were marked up using the “Track Changes” function.

Page 2 , Paragraph 6

“On the other hand, according to the World Health Organization, physical activity offers significant health benefits and mitigate health risks [27]. As a result, adults aged between 18 to 64 years should undertake at least 150 minutes of moderate or 75 minutes of vigorous-intensity physical activity, or an equivalent combination of them per week to prevent detrimental effects on health outcomes [27]. The role of physical activity still under investigation after COVID-19 infection, although”

  1. 2. There is too little information / study presented in the Introduction about PA and body mass, there are two very important variables in this study, please expand the chapter with new information on this level.

Page 2, Paragraph 5

Answer: We agree and thank you for your comment. We included this information in the manuscript (see below). All modifications were marked up using the “Track Changes” function.

“It is well established that other factors can directly affect autonomic function, such as obesity[21,22] and levels of physical activity [23,24]. Recent investigations indi-cates that obesity is associated with poor prognosis for SARS-CoV-2 infection, inten-sive care hospitalization and disease progression [25]. Additionally, each 1-unit in-crease in body mass index (BMI) might be related to a 12% increase in the risk of severe COVID-19. These alterations can be associated with chronic low-grade inflammation and impairments in immune cell activation of obese patients [25,26]. However, it is necessary to investigate if overweight and obesity can cause additional changes to the ANS after COVID-19 infection.

On the other hand, according to the World Health Organization, physical activity offers significant health benefits and mitigate health risks [27]. As a result, adults aged between 18 to 64 years should undertake at least 150 minutes of moderate or 75 minutes of vigorous-intensity physical activity, or an equivalent combination of them per week to prevent detrimental effects on health outcomes [27]. The role of physical activity still under investigation after COVID-19 infection, although sufficient physical activity decreases the prevalence of COVID-19-related hospitalization [28]. Therefore, it is important to understand if physical activity can play a protective role on the ANS after SARS-CoV-2 infection.”

  1. 3. Can you send us the proof of the ethics committee's agreement? I do not understand why this agreement consists so many numbers, i.e. ''38701820.0.0000.5402''

Answer: Thank you for your comment. The numbered presented is the Certificate of Presentation of Ethical Appreciation by the Ethical Review Board. We are sending the ethics committee's agreement in this review (see attached).

  1. 4. What happened or how the GT3X was used (ActiGraph, LLC, Pensacola, FL, USA) when the participants sleep? I do not find this information in the methodology, only when they took a bath, please fill in the necessary information.

Answer: We agree and thank you for your comment. This information was included (see below).

Page 4, paragraph 1

“Participants were instructed not to use the accelerometer while bathing, sleeping or performing water activities.”

  1. 5. References: please put in accordance with journal policy.

Answer: We agree and thank you for your comment. The citation system was corrected according instruction of the journal. All modifications marked up using the “Track Changes” function.

Reviewer 3 Report

The article entitled "Role Of Body Mass and Physical Activity in Autonomic Function Modulation on Post-COVID-19 Condition: An Observational Subanalysis Of Fit-COVID Study" is an interesting study for a major worldwide topic.The aim of this study was to analyze the autonomic function of young adults after mild-to-moderate infection with COVID-19 and to assess whether body mass index and levels of physical activity modulates autonomic function in participants with and without COVID-19.

I have some comments which should improve the manuscript:

Did the patients were vaccinated? this could be a confusing factor for the detection of antibodies. why did the authors not used the existence of previous positive PCR test for inclusion of patients. 

How patients with post-covid were defined? what were the criterion?

Provide the different symptoms of post-covid participants.

Provide the information of sexe and height in the two groups in tables.

In table 3, 4 and 5, provide the number of participants for each group.

1er sentence of the results, clarify that 57 subjects were evaluated but only 38 were analyzed, this is confusing.

the authors should clarify their classification for sedentary time: i.e. in methods as defined by over or below than 100 counts/min etc. (please provide a reference for this classification) but in statistical analysis groups of physical activity were defined as over or below 150 ( no references). Could the authors provide more information about this contrast, this is confusing.

is there some information/reporting about missing data? 

Please add that the low number of patients could be a limitation for the study. 

Author Response

  1. 1. Did the patients were vaccinated? this could be a confusing factor for the detection of antibodies.

Answer: We agree and thank you for your comment. We did not included vaccinated subjects in our study. We included this information in the manuscript (see below). All modifications were marked up using the “Track Changes” function.

Page 3, Paragraph 6

“The exclusion criteria were as follows: (1) presence of any chronic noncommuni-cable diseases, (2) smokers, (3) history of drug use, (4) medications such as anti-inflammatory drugs, antibiotics, and other known for their impact over the ANS, (5) individuals that were vaccinated for COVID-19 and (5) patients that received intensive care.”

  1. 2. Why did the authors not used the existence of previous positive PCR test for inclusion of patients?

Answer: Thank you for your comment. We did used the previous positive PCR test presented by participants in post-COVID group. To confirm that the control group was not previously infected by SARS-CoV-2 we conducted an IgM and IgG antibodies test for this group. This section was reformulated in the manuscript to clarify this information (see below). All modifications were marked up using the “Track Changes” function.

Page 3, Paragraph 5

We included male and female patients, aged 20-40 years after mild or moderate clinical COVID-19 with a previous positive PCR test and infection including slight clinical symptoms, fever, or respiratory symptoms and that were not admitted to the intensive care unit. Participants were recruited after a minimum of 15 and maximal of 180 days of diagnosis by positive PCR test [32]. An age-matched healthy control group that was negative for SARS-CoV-2 was also recruited. To screen for confirmed or probable pre-vious SARS-CoV-2 infection for the control group, a lateral flow test for IgM and IgG antibodies was conducted using internal anti-SARS-CoV-2 Immunoglobulin G (IgG) and Immunoglobulin M (IgM) ELISA kits.”

  1. 3. How patients with post-covid were defined? what were the criterion?

Answer: Thank you for your comment. The criteria for post-covid were defined as individuals with a previous positive PCR test and infection including slight clinical symptoms, fever, or respiratory symptoms and that were not admitted to the intensive care unit. Participants were recruited after a minimum of 15 and maximal of 180 days of diagnosis by positive PCR test. This section was reformulated in the manuscript to clarify this information. All modifications were marked up using the “Track Changes” function. See comment 2.

  1. 4. Provide the different symptoms of post-covid participants.

Answer: Thank you for your comment. The symptoms reported by participants during the infection are reported in the results section of the manuscript (see below).

Page 6, results section

“A mean of 5.75 ± 2.61 symptoms (clinical and respiratory) was noted. The main symp-toms reported and their respective prevalence were respiratory symptoms (n = 15; 75%), headache (n = 14; 70%), body pain (n = 14; 70%), anosmia (n = 10; 50%), ageusia (n = 9; 45%), fever (n = 6; 30%), and diarrhea (n = 5; 25%).”

  1. 5. Provide the information of sex and height in the two groups in tables.

Answer: Thank you for your comment. We included this information in Table 1 as suggested. All modifications were marked up using the “Track Changes” function in the manuscript.

  1. 6. In table 3, 4 and 5, provide the number of participants for each group.

Answer: Thank you for your comment. We included this information in Table 3,4 and 5 as suggested. All modifications were marked up using the “Track Changes” function in the manuscript.

  1. 7. 1er sentence of the results, clarify that 57 subjects were evaluated but only 38 were analyzed, this is confusing.

Answer: We agree and thank you for your comment. We performed the corrections as suggested (see below). All modifications marked up using the “Track Changes” function.

Page 5, Paragraph 1 of results section

“For this study, 57 subjects were evaluated. After 19 exclusions due to errors in HRV recordings, we analyzed 38 subjects.”

  1. The authors should clarify their classification for sedentary time: i.e. in methods as defined by over or below than 100 counts/min etc. (please provide a reference for this classification) but in statistical analysis groups of physical activity were defined as over or below 150 (no references). Could the authors provide more information about this contrast, this is confusing.

Answer: Thank you for your comment. It is important to clarify that sedentary time and physical inactivity are two different constructs. Sedentary behavior is defined as any waking behavior characterized by an energy expenditure ≤1.5 METs while in a sitting, reclining or lying; while the term physical inactive is used to describe people who are performing insufficient amounts of moderate- and vigorous-intensity activity (MVPA).1,2 In our study sedentary time was defined as values <100 counts/min and calculated by ActLife software (version 6.9.2, Pensacola, FL, USA) according to previous recommendations.3 Meanwhile, we considered individuals as physically active if they performed at least 150 minutes of MVPA per week, according to the World Health Organization 2020 guidelines. 4

References:

  1. Tremblay MS, Aubert S, Barnes JD, Saunders TJ, Carson V, Latimer-Cheung AE, et al. Sedentary Behavior Research Network (SBRN) - Terminology Consensus Project process and outcome. The international journal of behavioral nutrition and physical activity. 2017;14(1):75-.
  2. van der Ploeg HP, Hillsdon M. Is sedentary behaviour just physical inactivity by another name? Int J Behav Nutr Phys Act. 2017;14(1):142.
  3. Troiano, R.P.; Berrigan, D.; Dodd, K.W.; Mâsse, L.C.; Tilert, T.; McDowell, M. Physical activity in the United States measured by accelerometer. Medicine and science in sports and exercise 2008, 40, 181-188.
  4. Bull, F.C.; Al-Ansari, S.S.; Biddle, S.; Borodulin, K.; Buman, M.P.; Cardon, G.; Carty, C.; Chaput, J.P.; Chastin, S.; Chou, R.; et al. World Health Organization 2020 guidelines on physical activity and sedentary behaviour. British journal of sports medicine 2020, 54, 1451-1462.
  5. Is there some information/reporting about missing data?

Answer: Thank you for your comment. The exclusions in our study were due to the errors due to errors in HRV recordings, therefore we did not include this individuals in the final analysis. The remaining  38 individuals presented complete data and were included for final analysis. This information was included in the manuscript (see below). All modifications marked up using the “Track Changes” function.

Page 23, Results, Paragraph 1

“For this study, 57 subjects were evaluated. After 19 exclusions due to errors in HRV recordings, 38 subjects with complete data were included in the final analysis.”

  1. Please add that the low number of patients could be a limitation for the study.

Answer: We agree and thank you for your comment. We performed the corrections as suggested (see below). All modifications marked up using the “Track Changes” function.

Page 18, Paragraph 6

“The main limitations to our investigation were (1) the cross-sectional nature of the study and the small sample size (n=57). This is of particular concern because our primary outcome measure (HRV) is characterized by high interindividual variability.”

Round 2

Reviewer 1 Report

The authors made several changes to the paper. Unfortunately, a ritical issues remain that prevent its publication. The number of subjects (57) is very small for a cross-sectional study. Usually hundreds of patients are evaluated in this type of study. The division into groups was designed in such a way that an assessed group would consist of only few individuals. Splitting the sample in this way makes the results scarcely convincing and does not allow scientifically reliable conclusions to be drawn. 

Author Response

Thank you for your comment. Indeed the sample size is an important limitation of our study and this was better pointed out at our discussion section. 

Reviewer 3 Report

Acceptance

Author Response

Thank you for the opportunity to have our paper improved.